# Influence of Coal-Fired Fly Ash on Measurement Error of NO_2_ Electrochemical Sensors

**DOI:** 10.3390/s24030900

**Published:** 2024-01-30

**Authors:** Wei Chen, Hui Zhou, Shijing Wu, Dongmei Liao

**Affiliations:** 1The Institute of Technological Sciences, Wuhan University, Wuhan 430072, China; chenwei2020@whu.edu.cn (W.C.); ldm@whu.edu.cn (D.L.); 2School of Power and Mechanical Engineering, Wuhan University, Wuhan 430072, China; 2022282080093@whu.edu.cn

**Keywords:** electrochemical sensors, NO_2_ sensors, coal-fired fly ash, measurement error

## Abstract

To overcome the limitations of NO_2_ electrochemical sensors, including their inaccurate measurements and short working life, when used around coal-fired power plants, we investigated the influence of coal-fired fly ash deposition on the measurement error of NO_2_ electrochemical sensors through experimental tests. The morphological characteristics and pellet diameter distribution of coal-fired fly ash pellets were determined via scanning electron microscopy. The sedimentation velocity of coal-fired fly ash pellets in the air was determined through theoretical calculations of aerodynamics and hydrodynamics. Additionally, the effect of the deposition of coal-fired fly ash on the measurement error of NO_2_ electrochemical sensors was determined through experimental tests. The test results show that the minimum and maximum measurement errors of the NO_2_ electrochemical gas sensor were 8.015% and 30.35%, respectively, after a deposition duration of 30 days with 30 mg/m^3^ coal-fired fly ash. This demonstrates that coal-fired fly ash deposition is the cause of the inaccurate measurements and short working life of these sensors. Coal-fired fly ash causes a decrease in the gas diffusion area of the sensor and the diffusion coefficient, thus increasing the sensor measurement error.

## 1. Introduction

In the first three quarters of 2023, China’s cumulative installed power generation capacity was approximately 2.79 billion kW, which is a year-on-year increase of 12.3%. Of this, the installed capacity of thermal power was 1.37 billion kW, accounting for 49.18% of the total installed capacity [1], indicating that China’s electricity was mainly produced from thermal power. Thermal power plants generate electricity by burning coal, which produces large quantities of soot, coal-fired fly ash [2], NO, NO_2_, and other pollutants and gases during combustion [3,4,5]. NO and NO_2_ not only form photochemical smog and acid rain and destroy ozone [6,7], they also harm the human body by affecting the central nervous system and damaging the heart and respiratory system [8,9]. The lethal concentrations of NO in humans are 2500 ppm (exposure time 30 min) and 10,000 ppm (exposure time 5 min); the lethal concentrations of NO_2_ in humans are 500 ppm (exposure time 30 min) and 5000 ppm (exposure time 5 min) [10,11]. Therefore, pollutant gases such as NO and NO_2_ in the air near coal-fired power plants must be monitored.

The NOx gas concentration in the air pollutant produced by thermal power plants is mainly calculated using the concentration of NO_2_ gas [12]. As NO_2_ gas is more toxic than NO gas, NO_2_ gas was selected as the object in this study. The methods commonly used for detecting NO_2_ gas include the metal–oxide semiconductor sensor detection [13,14], electrochemical gas sensor detection [15,16,17], and N-(1-naphthyle) ethylenediamine dihydrochloride spectrophotometric [18] methods. The limitations of metal–oxide semiconductor sensors include their poor durability, low accuracy, and low gas selectivity. The N-(1-Naphthyl)ethylenediamine dihydrochloride spectrophotometric method can be used to accurately measure NO_2_ gas concentrations, but the on-site manual sampling and measurement of NO_2_ gas are required: remote measurement is impossible. Conversely, electrochemical gas sensors are cheap, simple in structure, small, energy-efficient, and stable, and provide a fast response, strong linearity, high sensitivity, and high selectivity; as such, they are widely used for safety monitoring, environmental protection, food safety, and in industry [19,20]. Currently, studies on electrochemical sensors have mainly focused on the structure of the sensor, the material of the working electrode, and the material of the electrolyte [21,22,23,24,25]; however, few studies have been conducted on the engineering applications of electrochemical sensors. In practical engineering applications, the electrochemical sensors that are used to monitor the concentration of pollutants emitted by coal-fired power plants suffer from a low measurement accuracy and short service life.

Studies in this field on fly ash environments are scarce. Ghenadii Korotcenkov [26] showed that when electrochemical gas sensors are used in dusty areas, the diffusion membrane becomes clogged or coated, and the normal supply of the signal gas may be cut off, leading to sensor failure. However, the effects of dust clogging on the measurement error of a sensor have not been described. 

Therefore, the focus of this study was a NO_2_ electrochemical gas sensor, and the effect of coal-fired fly ash on the measurement error of this sensor was investigated through experiments. Suggestions are provided in this paper for the use and maintenance of equipment using electrochemical sensors in coal-fired power plant areas. Firstly, the coal-fired fly ash specimens were analyzed using scanning electron microscopy to determine the morphological characteristics and pellet diameter distribution of the coal-fired fly ash pellets. Secondly, according to the theoretical aerodynamics and hydrodynamics calculations, the sedimentation velocity of coal-fired fly ash pellets in air and the deposited quantities of coal-fired fly ash at various time points were determined. Finally, the impact of coal-fired fly ash on the measurement error of NO_2_ electrochemical sensors was determined through experimental testing.

## 2. Materials and Methods

### 2.1. Experimental Materials

#### 2.1.1. Experimental Equipment

A TESCAN MIRA3 field-emission scanning electron microscope (TESCAN GROUP, Brno, Czech Republic) was used to analyze the morphological characteristics, elemental composition, and pellet diameter distribution of the coal-fired fly ash, which had a resolution of 2.0 nm at 30 keV in low-vacuum mode and 1.2 nm at 30 keV or 2.5 nm at 3 keV in high-vacuum mode.

#### 2.1.2. NO_2_ Electrochemical Gas Sensor

The Chinese domestic manufacturers of these sensors include Hanwei Technology Group Co., Ltd. (Zhengzhou, China), Cube Sensor and Instrument Co., Ltd. (Wuhan, China), Weihai Jingxun Tongtong Electronic Technology Co., Ltd. (Weihai, China), and Guangzhou Aosong Electronics Co., Ltd. (Guangzhou, China). The foreign manufacturers are Honeywell (Morristown, NJ, USA), Alphasense (Essex, UK), CityTech (London, UK), and Figaro (Minoo city, Osaka, Japan).

Comparing the specifications of the NO_2_ electrochemical sensors manufactured between Chinese and foreign manufacturers, we found that the NO_2_ electrochemical gas sensors from these manufacturers have the same operating principle and similar construction. Alphasense’s electrochemical sensors were the most precise and sensitive, and the company had the largest market share, so we purchased the NO_2_-B43F model of the NO_2_ electrochemical sensor from Alphasense for our experimental tests.

The appearance and dimensions of the NO_2_ electrochemical gas sensor are presented in Figure 1. The diameter of the gas diffusion membrane on the sensor surface was 19 mm, with an area of 283.5 × 10^−6^ m^2^, and the thickness of the membrane was 0.28 µm. The material was made of porous polytetrafluoroethylene with a porosity of 49% and a pore size of 12–16 µm. The performance parameters of the NO_2_ electrochemical gas sensor are listed in Table 1 [27].

Structurally, the NO_2_ electrochemical gas sensor included a gas diffusion layer, wetting filters, an electrolyte solution, a working electrode, a reference electrode, and a counter electrode, as illustrated in Figure 2. The gas sensing mechanism of the NO_2_ electrochemical gas sensor is shown in Figure 3.

The two transport paths through which NO_2_ gas can diffuse into the interior of the sensor are as follows: In the first path, NO_2_ gas passes directly through the electrolyte solution (ionic cluster) and undergoes an electrochemical reaction in the b1 plane, which satisfies the three-phase point (working electrode–electrolyte solution–gas). In the second path, NO_2_ gas first permeates the wetting filters membrane to the boundary of the electrolyte solution (ionic cluster) and then diffuses toward the surface of the working electrode, where an electrochemical reaction occurs in the b2 plane. The gas transport is much faster for path 1 than for path 2, so path 1 almost completely determines the value of the signal produced by the sensor. When NO_2_ gas with a concentration of C_0_ is transported along path 1, the gas concentration changes to C_1_ after the gas passes through the electrolyte solution (ionic clusters), owing to a small amount of gas dissolving in the electrolyte solution. Then, the gas is transferred to the working electrode, and an electrochemical reaction occurs, resulting in the gas concentration changing to C_2_. As the electrochemical reaction continues, the concentration of the gas changes to C_3_, and the gas diffuses further into the electrolyte solution, causing the concentration to continue to decrease.

NO_2_ gas diffuses through the gas diffusion layer to the working electrode, where an oxidation reaction occurs [28]:(1)$$N\;O_{2}+H_{2}O\to N\;O_{2}^{-}+2\;H^{+}+e$$

A diffusion current i is generated at the working electrode. In certain working situations, Faraday’s constant F, electron transfer number Z, diffusion coefficient D, gas diffusion area S, and thickness of diffusion layer δ are all constants, so that the diffusion current i is proportional to the NO_2_ gas concentration (ρ) within a certain range [29].
(2)$$i=\frac{Z\cdot F\cdot S\cdot D}{\delta }\times \rho$$

#### 2.1.3. Coal-Fired Fly Ash Samples

The coal-fired fly ash samples were obtained from an ash hopper of the boiler dust collector of a thermal power plant, as shown in Figure 4. Clean, transparent polyethylene plastic bags were used to sample the coal-fired fly ash, which ensured that the coal-fired fly ash samples were not contaminated, at three different time periods (10:00 to 11:00, 16:00 to 17:00, and 21:00 to 22:00). A total of 3 kg of fly ash was collected at each time period to ensure the homogeneity of the coal-fired fly ash samples.

#### 2.1.4. Experimental Gases

The experimental gas was N_2_ standard gas. High-purity nitrogen with 99.999% purity.

Standard NO_2_ gas at concentrations of 1000 ppm. The standard NO_2_ gas was diluted to 10 ppm, 20 ppm, 30 ppm, 40 ppm, 50 ppm, 60 ppm, 70 ppm, 80 ppm, 90 ppm, and 100 ppm using gas flow meter and N_2_ standard gas.

### 2.2. Experimental Methods

#### 2.2.1. Drying of Coal-Fired Fly Ash Samples

The coal-fired fly ash samples were dried prior to morphology scanning. A total of 2 kg of coal-fired fly ash samples was evenly placed on a tray, which was placed into a drying oven at 120 °C [30]. After 120 min of drying, the weighing results showed that no change occurred in the mass of the coal-fired fly ash samples, indicating that the drying was complete.

#### 2.2.2. Experimental Tests of Measurement Error of NO_2_ Electrochemical Gas Sensors

The experimental test device consisted of a test box body, a test box cover, an organic rubber sealing ring, connecting bolts, an inlet valve, an outlet valve, and a sensor support base, as illustrated in Figure 5. The test box body and cover were made of plexiglass. The inner diameter of the test box was 270 mm, the height was 300 mm, and the wall thickness was 15 mm. The test box body and cover were sealed with an organic rubber sealing ring, which was fastened using 6 connecting bolts, evenly distributed at an interval of 60 degrees to ensure the airtightness of the test box.

The dimensions of the test device were chosen for the following reasons: The sensors needed to be placed inside the test device, so the internal dimensions of the test device needed to be spacious enough for mounting the sensors. Additionally, if the test device size was too large, such as an inner diameter of 1000 mm and a height of 800 mm, the test box would have been heavy and difficult to handle. Finally, more NO_2_ gas would require the filling of a large test device, which would increase the difficulty of treating the experimental exhaust and would risk NO_2_ gas leakage.

The experimental test of the NO_2_ electrochemical gas sensor’s measurement errors involved the following steps:

Firstly, two NO_2_ electrochemical gas sensors were placed in the experimental test device (sensor No. 1 was a blank sensor, and sensor No. 2 was an experimental sensor). Then, coal-fired fly ash deposited for 30 days was placed on the surface of the diffusion membrane of sensor No. 2.

Secondly, the experimental test device was sealed using bolts, the inlet valve was opened, and standard NO_2_ gas at a concentration of 10 ppm was allowed to flow into the experimental test device.

Thirdly, the NO_2_ electrochemical gas sensor measurements were recorded, and the test was repeated three times. The average of the three recordings was used to calculate the sensor measurement error.

Fourthly, standard NO_2_ gas at a concentration of 20 ppm (or 30, 40, 50, 60, 70, 80, 90, or 100 ppm) was then introduced into the experimental test device, and the measurements of the NO_2_ electrochemical gas sensor were recorded.

Fifthly, the coal-fired fly ash deposited for 60 days (or 90, 120, 150, or 180 days) was placed on the surface of the diffusion membrane of sensor No. 2, and standard NO_2_ gas at a concentration of 10, 20, 30, 40, 50, 60, 70, 80, 90, or 100 ppm was passed through the membrane. The test was repeated three times, and the measurements of the NO_2_ electrochemical gas sensor were recorded.

## 3. Results

### 3.1. Morphological Characteristics, Elemental Composition, and Pellet Diameter Distribution of Coal-Fired Fly Ash

#### 3.1.1. Morphological Characteristics of Coal-Fired Fly Ash

The dried fly ash samples were analyzed via scanning electron microscopy. The morphological characteristics of the coal-fired fly ash are depicted in Figure 6. The coal-fired fly ash pellets appeared in the form of regular spheres with different diameters. The surface of the spherical pellets was smooth, the boundary between the pellets was well defined, and some tiny pellets adhered to large pellets.

#### 3.1.2. Elemental Composition and Proportion of Coal-Fired Fly Ash

The elemental composition of the coal-fired fly ash is shown in Figure 7, and the elemental ratios are listed in Table 2. The main elements that comprised the coal-fired fly ash were O, Si, Al, C, Fe, K, Ca, and Na; the main compounds were SiO_2_, Al_2_O_3_, CaO, and Fe_2_O_3_.

#### 3.1.3. Pellet Diameter Distribution of Coal-Fired Fly Ash

The pellet diameter statistics and pellet diameter distribution of the coal-fired fly ash are described in Figure 8 and Table 3. The range of the pellet diameter distribution was 1 to 21 µm with a minimum of 1.65 µm, a maximum of 20.07 µm, and an average of 5.48 µm. Moreover, 92.18% of the pellets were less than 11 µm in diameter, whereas 79.01% of the pellets were greater than 2.5 µm in diameter.

According to the statistical results, the pellet diameter of the coal-fired fly ash was mainly 2.5–9.83 μm, categorizing it as particulate matter PM2.5 and PM10 suspended in the atmosphere environment. Because larger particles such as PM50 and PM100 tend to settle quickly and are only suspended in the atmosphere for a short time, the main particulate pollutants in the atmosphere are PM2.5 and PM10. The emissions from coal combustion in thermal power plants are an important cause of air pollution.

### 3.2. Derivation of Sedimentation Velocity of Coal-Fired Fly Ash (Appendix A: Interpretation of Formula Symbols)

The air temperature and humidity, wind speed and direction, and airflow state (laminar and turbulent flow) can affect the sedimentation process of coal-fired fly ash. Simulating field air conditions in a laboratory and conducting tests at a coal-fired power plant are difficult. Therefore, the air conditions in the laboratory were set to a relatively ideal state (20 °C, 50% RH, wind speed of 0 m/s, and laminar flow), which simplified the sedimentation process, facilitated the derivation of the sedimentation velocity, and enabled the calculation of the quantities of coal-fired fly ash on the basis of fluid mechanics and aerodynamics. The sensor’s working temperature was −30 to 40 °C, and the working humidity was 15% to 85% RH. The air conditions set up in the laboratory met the requirements of the NO_2_ electrochemical sensor.

The sedimentation process and force values of the coal-fired fly ash pellets in air in the laboratory are illustrated in Figure 9. The sedimentation process comprised three stages, namely, stationary, accelerated sedimentation, and uniform sedimentation, according to the sedimentation velocity of the coal-fired fly ash pellet.

In the stationary state stage, the coal-fired fly ash pellet is only subjected to the force of gravity $G_{p}$ and air buoyancy $F_{f}$, and the sedimentation velocity of the coal-fired fly ash pellet $v_{p}$ is zero.
(3)$$G_{p}=m_{p}g=\rho_{p}V_{p}g=\frac{1}{6}\pi d_{p}^{3}\rho_{p}g$$
(4)$$F_{f}=\rho_{a}V_{p}g=\frac{1}{6}\pi d_{p}^{3}\rho_{a}g$$
(5)$$F_{s}=G_{p}-F_{f}=\frac{1}{6}\pi d_{p}^{3}\left(\rho_{p}-\rho_{a}\right)g$$
when the air buoyancy $F_{f}$ is less than gravity $G_{p}$, the coal-fired fly ash pellets start accelerating and progressing to the accelerated sedimentation stage. At this time, air resistance $F_{r}$ is generated. The kinematic formula of the coal-fired fly ash pellets is
(6)$$F_{s}-F_{r}=m_{p}a_{p}=m_{p}\frac{dv_{p}}{dt}$$

The air resistance $F_{r}$ can be calculated as follows [31]:(7)$$F_{r}=\frac{1}{2}C_{a}A_{p}\rho_{a}{v_{p}}^{2}$$

The air resistance of the coal-fired fly ash pellets with a regular spherical shape can be calculated as
(8)$$F_{r}=\frac{1}{2}C_{a}\frac{\pi {d_{p}}^{2}}{4}\rho_{a}{v_{p}}^{2}=C_{a}\frac{\pi {d_{p}}^{2}\rho_{a}{v_{p}}^{2}}{8}$$

The expression for the sedimentation velocity of the coal-fired fly ash pellets can be obtained by substituting Formulas (5) and (8) into Formula (6):(9)$$\frac{dv_{p}}{dt}=\frac{\left(\rho_{p}-\rho_{a}\right)g}{\rho_{p}}-\frac{3C_{a}\rho_{a}}{4d_{p}\rho_{p}}{v_{p}}^{2}$$

The air resistance $F_{r}$ increases with the increase in the velocity. When the air resistance $F_{r}$ is equivalent to the total force $F_{s}$, the sedimentation velocity of the coal-fired fly ash pellet reaches its maximum value $v_{\mathrm{pmax}}$. Then, the coal-fired fly ash pellets start falling at a uniform velocity and progress to the uniform sedimentation stage, and the acceleration is zero.
(10)$$\frac{dv_{\mathrm{pmax}}}{dt}=\frac{\left(\rho_{p}-\rho_{a}\right)g}{\rho_{p}}-\frac{3C_{a}\rho_{a}}{4d_{p}\rho_{p}}{v_{\mathrm{pmax}}}^{2}=0$$
(11)$$v_{p\max}=\sqrt{\frac{4d_{p}\left(\rho_{p}-\rho_{a}\right)g}{3C_{a}\rho_{a}}}$$

The air resistance coefficient $C_{a}$ is a function of the Reynolds number $Re_{p}$ [32], which can be calculated as
(12)$$C_{a}=\frac{k}{R{e_{p}}^{m}}$$
when the airflow is laminar, $Re_{p}\leq 1$, m=1, and k=24. The air resistance coefficient $C_{a}$ can then be calculated as
(13)$$C_{a}=\frac{24}{Re_{p}}=\frac{24\mu_{a}}{\rho_{a}v_{p}d_{p}}$$

The expression for the maximum sedimentation velocity $v_{\mathrm{pmax}}$ can be obtained by substituting Formula (13) into Formula (11):(14)$$v_{p\max}=\frac{\left(\rho_{p}-\rho_{a}\right)g{d_{p}}^{2}}{18\mu_{a}}$$

The tap density of the coal-fired fly ash pellet $\rho_{p}$ is 1200 kg/m^3^, according to calculations [33]. Referring to the standard atmosphere table [34], the density of air $\rho_{a}$ is 1.205 kg/m^3^ and the aerodynamic viscosity $\mu_{a}$ is 1.810 × 10^−5^ Pa·s when the air temperature is 20 °C.

The range of the pellet diameter distribution was 1 to 21 µm, with a minimum of 1.65 µm, a maximum of 20.07 µm, and an average of 5.48 µm. Therefore, the range of $v_{\mathrm{pmax}}$ was
(15)$$\begin{array}{l} {v_{p}}_{\max-\min}=v_{d_{p}=1.65\mu m}=0.98\times 10^{-4}\;m/s\\ {v_{p}}_{\max-\max}=v_{d_{p}=20.07\mu m}=14.52\times 10^{-3}\;m/s \end{array}$$

As 92.18% of the pellet was less than 11 µm in diameter, and 79.01% of the pellet was more than 2.5 µm in diameter, the sedimentation velocity $v_{p}$ was calculated based on the average of 5.48 µm.
(16)$$v_{p}=v_{d_{p}=5.48\mu m}=10.83\times 10^{-4}\;m/s$$

The theoretical calculation method enables the modeling of the sedimentation process of the coal-fired fly ash pellet, which is convenient for analyzing its movement state. However, the model does not consider the effects of the wind speed and direction or airflow state. Despite these shortcomings, this method is valid for measuring the sedimentation velocity of coal-fired fly ash pellets in laboratory air environments and has important engineering application value in the on-site analysis of these pellets at coal-fired power plants.

### 3.3. Experimental Test Results of Measurement Error of NO_2_ Electrochemical Sensor

The Emission Standard of Air Pollutants for Thermal Power Plants [12] in China establishes a limit of 30 mg/m^3^ for smoke and dust emissions from coal boilers, so we selected this limit for our study. 

This concentration was chosen for the experimental tests for two reasons. Firstly, the fly ash concentration at the coal-fired power plant was too low to quickly cause a sensor measurement error. We installed a fly ash concentration monitoring device 100 m from the coal-fired power plant and found that the fly ash concentration was 1 mg/m^3^. Initially, we chose a fly ash concentration of 1 mg/m^3^ for the experimental tests and found that the low fly ash concentration resulted in zero deposition, preventing the completion of the experimental tests. Therefore, we ultimately selected 30 mg/m^3^ for the experimental testing. Secondly, 30 mg/m^3^ can realize time acceleration effects. In the laboratory atmosphere, 1 day’s deposition at 30 mg/m^3^ was assumed to be equal to 30 days’ deposition at 1 mg/m^3^.

A cube (1 m × 1 m × 1 m) was set up in the laboratory, and then 30 mg/m^3^ coal-fired fly ash was introduced from the top surface of the cube. The fly ash started to settle from the top surface onto the bottom surface of the cube. According to formula (16), we calculated that the time required for the coal-fired fly ash to settle on the bottom surface of the cube was 923.4 s (about 15.4 min). After the coal-fired fly ash completely settled on the bottom surface, as the bottom area was 1 m^2^, the mass of coal-fired fly ash that settled on the bottom surface was 30 mg. 

The NO_2_ electrochemical sensor was placed at the bottom of the cube, and 30 mg/m^3^ coal-fired fly ash was also introduced from the top surface. Because the area of the sensor’s gas diffusion membrane was 283.5 × 10^−6^ m^2^, the amount of coal-fired fly ash deposited on the surface of the membrane was calculated to be 8.5 × 10^−3^ mg. The amount of coal-fired fly ash deposited on the surface of the sensor gas diffusion membrane at different times was obtained by changing the time at which the fly ash was introduced.

Table 4 shows the quantities of 30 mg/m^3^ coal-fired fly ash deposited on the gas diffusion membrane after various deposition durations.

The NO_2_ electrochemical sensor was placed in the test box, and the coal-fired fly ash samples, which were deposited for 30, 60, 90, 120, 150, or 180 days, were uniformly placed on the gas diffusion membrane of sensor No. 1. Afterward, NO_2_ gas (10, 20, 30, 40, 50, 60, 70, 80, 90, or 100 ppm) was introduced to the test box. The experimental measurement and sensor measurement error values are presented in Table 5 and Table 6. The data handling results are illustrated in Figure 10.

Analyzing the experimental measurements, we found that the minimum and maximum measurement errors of the NO_2_ electrochemical sensor, respectively, were 8.015% and 30.35% for a deposition duration of 30 days, 11.536% and 49.8% at 60 days, 13.441% and 63.18% at 90 days, 20.768% and 67.68% at 120 days, 24.03% and 78.18% at 150 days, and 30.78% and 89.49% at 180 days. The measurement error of the NO_2_ electrochemical sensor increased with the increasing coal-fired fly ash deposition time.

The measurement error was calculated as follows: NO_2_ gas concentration value minus sensor-measured value, multiplied by 100%. For example, for a deposition time of 30 days, the measurement error for 10 ppm NO_2_ would be 30.35% = (10 − 6.965) × 100%.

This occurs because the deposition of coal-fired fly ash on the gas diffusion membrane prevents the gas from entering the sensor, leading to measurement errors. The pellet diameter ranged from 1 to 21 µm, and the gas diffusion membrane had 49% porosity and a 12–16 µm pore diameter. As coal-fired fly ash is deposited on a gas diffusion membrane, the membrane blocks the coal-fired fly ash pellets with a diameter of less than the pore diameter from entering the sensor via electrostatic adsorption, inertial collision, and diffusion collision [35]. This results in a monolayer of fly ash forming on the gas diffusion membrane, blocking some of its pores, resulting in a decrease in S in Formula (2). This produces a negative measurement error, i.e., sensor-measured values that are lower than the actual NO_2_ concentration. Additionally, the thickness of the coal-fired fly ash deposition layer increases over time. The single layer of fly ash gradually becomes multiple layers, resulting in a decrease in D in Formula (2), which leads to a reduction in the measured values and an increase in the measurement error.

### 3.4. Impact of Humidity on Measurement Errors of NO_2_ Electrochemical Gas Sensors 

A NO_2_ electrochemical gas sensor was placed in a humidity control device for humidity control, as shown in Figure 11. The humidity was 25% RH, 50% RH, 75% RH, or 98% RH, and each humidity level was maintained for 12 h. The results of the effects of the humidity on the measurement errors of the NO_2_ electrochemical gas sensor are shown in Table 7.

The results of the tests showed that the humidity had minimal effects on the measurement error: the higher the humidity, the smaller the measurement error. The average measurement error was 2.86% at 25% RH, 2.45% at 50% RH, 1.59% at 75% RH, and 1.08% at 98%. Therefore, the humidity essentially has a negligible effect on the sensor’s measurement error. The humidity only introduces large measurement errors if the sensor is placed in an environment with greater than 95% or less than 10% humidity for a long time. Under these conditions, the electrolyte solution inside the sensor is diluted or dries out, respectively.

### 3.5. Impact of Humidity-Coal Fly Ash Deposition on Measurement Errors of NO_2_ Electrochemical Gas Sensors 

The sensors on which coal fly ash were deposited were placed in the humidity control device; the humidity levels were set to 25% RH, 50% RH, 75% RH, or 98% RH and maintained for 12 h. The test results of the effects of the humidity with coal fly ash deposition on the measurement errors of the NO_2_ electrochemical gas sensor are shown in Table 8.

The test results showed that the measurement error of the NO_2_ electrochemical gas sensor affected by the humidity and coal fly ash deposition was larger than that of the dried coal fly ash deposition, where the higher the humidity, the larger the measurement error of the sensor. In humid environments, dried coal fly ash would absorb water vapor from the air. When NO_2_ gas passes through the inlet membrane and enters the sensor, some part of the NO_2_ gas is absorbed by the moisture in the deposited coal fly ash, which leads to a decrease in the concentration of the gas and produces a measurement error. As such, the higher the humidity, the more NO_2_ gas is absorbed by the moisture, and the larger the measurement error.

## 4. Discussion

In this study, we found that coal-fired fly ash deposition reduced the diffusion coefficient D and gas diffusion area S of a sensor. The measurement error increased with the deposition time, whereas the measurement errors decreased as the gas concentration increased.

The limitation of our methodology is that the concentration of the coal-fired fly ash used in testing was higher than that measured in the field, leading to the quick failure of the sensor, which does not mimic the situation in practical applications. We installed a fly ash concentration monitoring device 100 m from a coal-fired power plant and monitored a fly ash concentration of 1 mg/m^3^. The device worked normally for 6 months before it failed.

We also discovered that removing the deposited fly ash from the gas diffusion membrane could reduce the measurement error. An effective method to address the decrease in the sensitivity of electrochemical sensors due to the deposition of coal-fired fly ash would involve preventing the coal-fired fly ash from being deposited on the sensor surface. Therefore, we recommend that a filter membrane be installed on the outside of the sensor’s equipment case (rather than on the outside surface of the sensor) to prevent coal-fired fly ash from entering the sensor. For example, a layer of filter membrane (polytetrafluoroethylene with a porosity of 50%, a pore size of 2–5 μm, and a thickness of 0.3 mm) could be installed at the air inlet of the equipment box, as shown in Figure 12. In addition, the periodic removal of deposited fly ash could extend the sensor’s working life and reduce the measurement error.

## Figures and Tables

**Figure 1 sensors-24-00900-f001:**
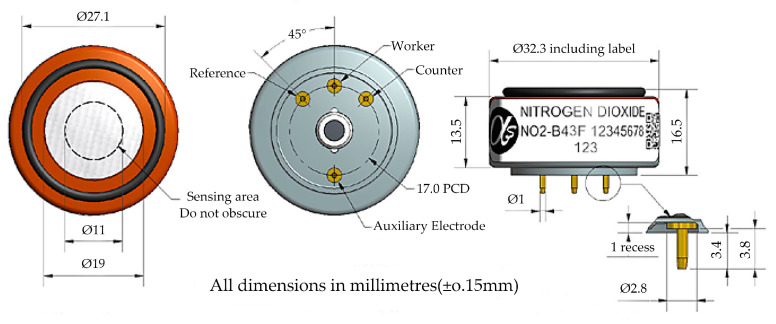
Appearance and dimensions of the NO_2_ electrochemical gas sensor.

**Figure 2 sensors-24-00900-f002:**
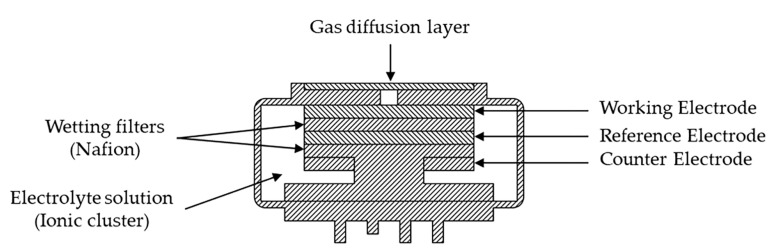
Structure diagram of NO_2_ electrochemical sensor.

**Figure 3 sensors-24-00900-f003:**
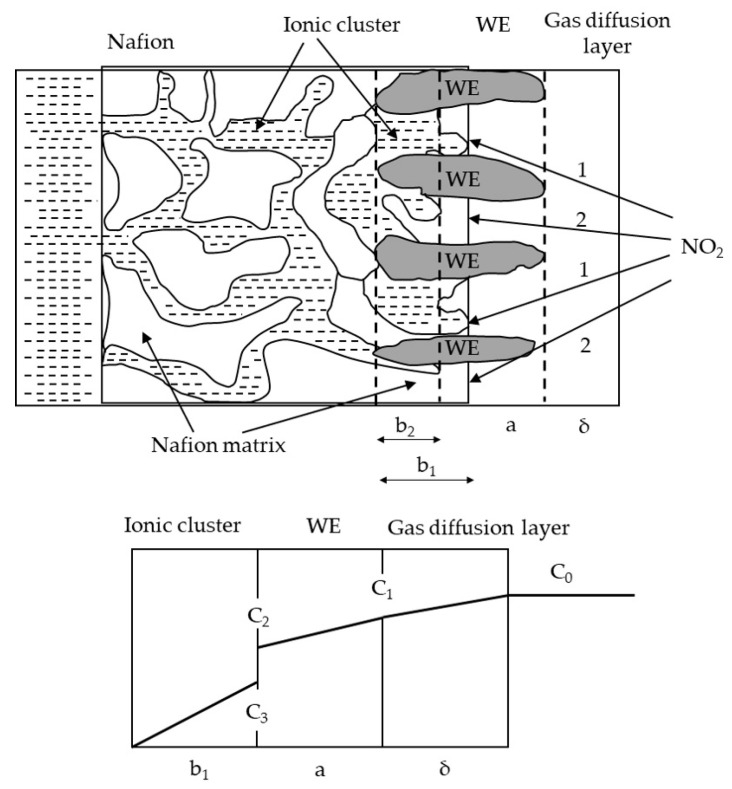
The transport pathways of and concentration changes in NO_2_ gas.

**Figure 4 sensors-24-00900-f004:**
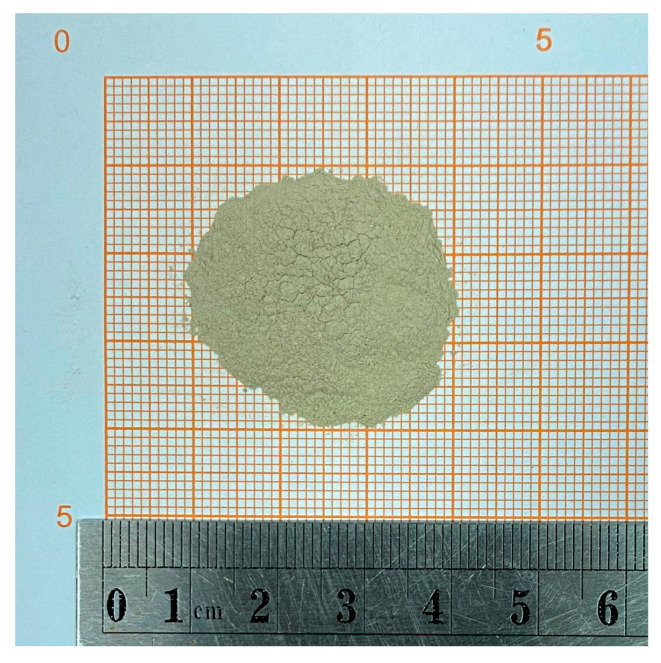
Coal-fired fly ash samples.

**Figure 5 sensors-24-00900-f005:**
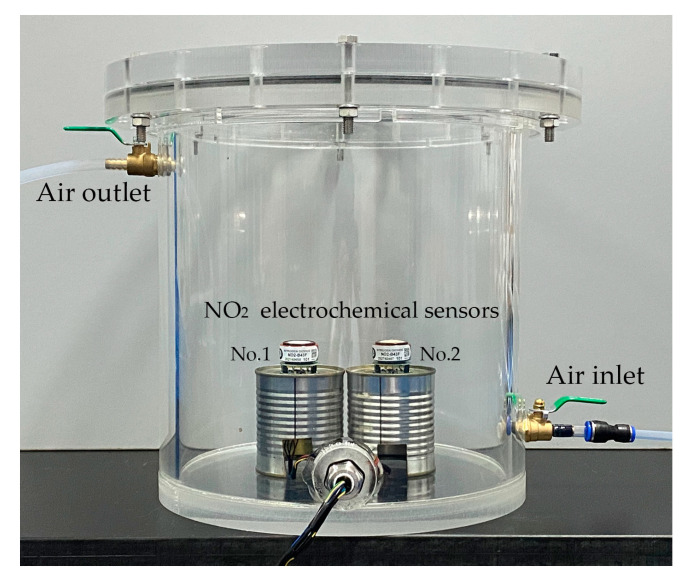
Experimental test device used for NO_2_ electrochemical sensors.

**Figure 6 sensors-24-00900-f006:**
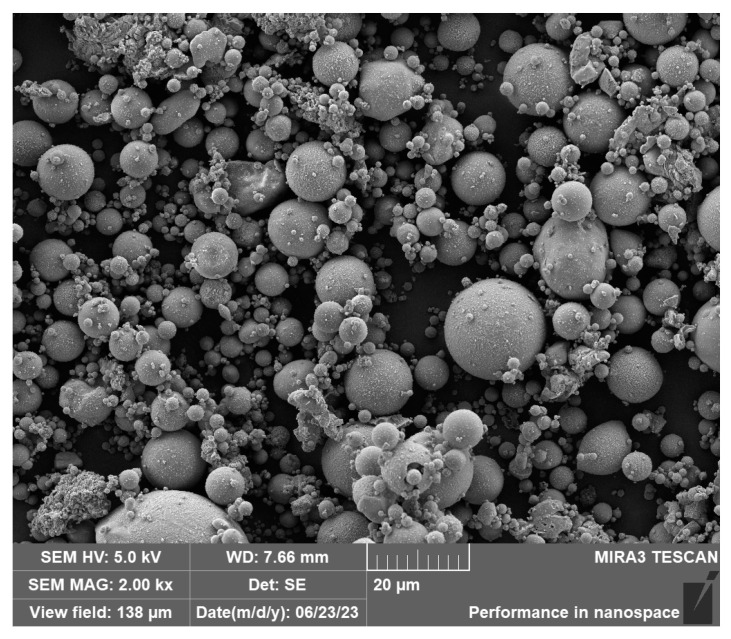
MIRA3 scanning result: morphological characteristics of coal-fired fly ash.

**Figure 7 sensors-24-00900-f007:**
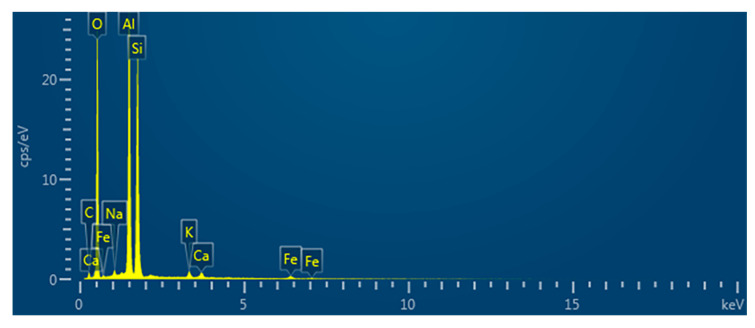
The elemental composition of coal-fired fly ash.

**Figure 8 sensors-24-00900-f008:**
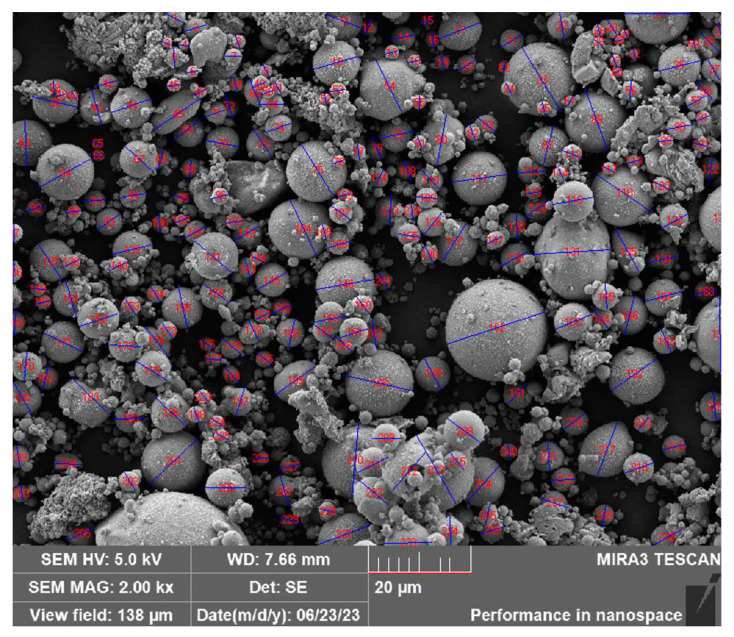
Pellet diameter distribution of coal-fired fly ash determined using Nano Measurer 1.2 software.

**Figure 9 sensors-24-00900-f009:**
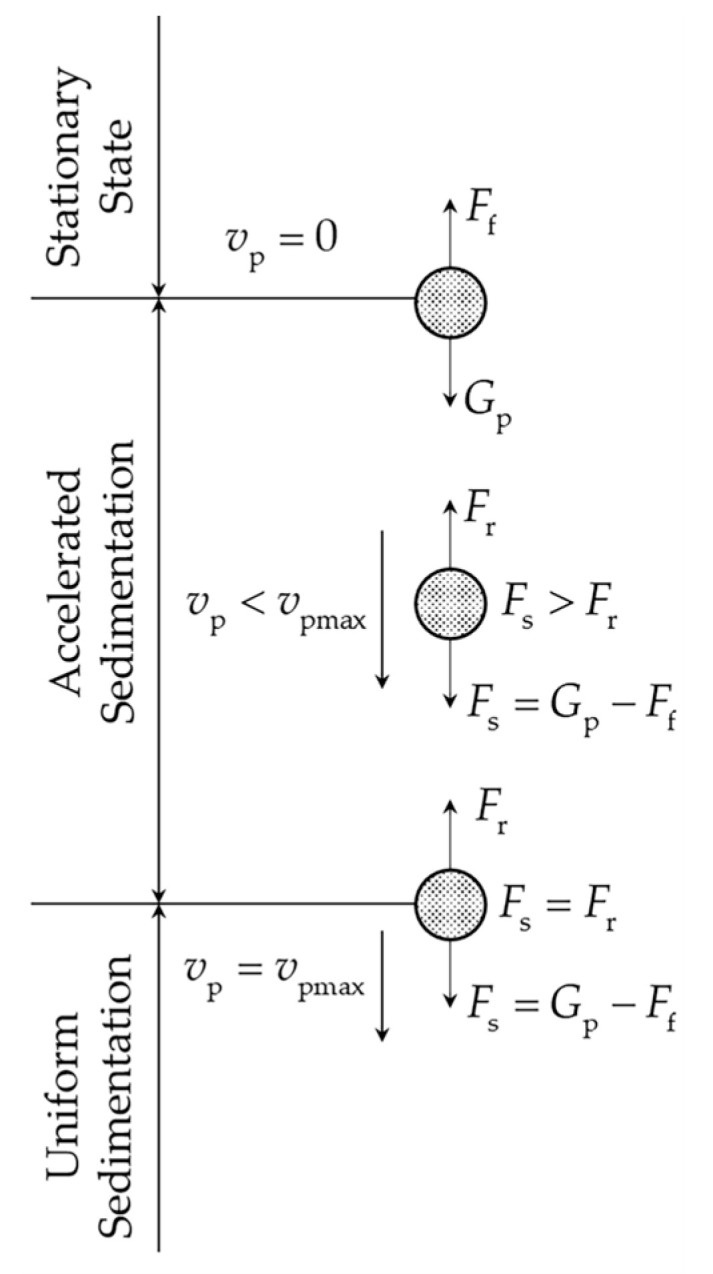
The sedimentation process and force values of coal-fired fly ash pellet.

**Figure 10 sensors-24-00900-f010:**
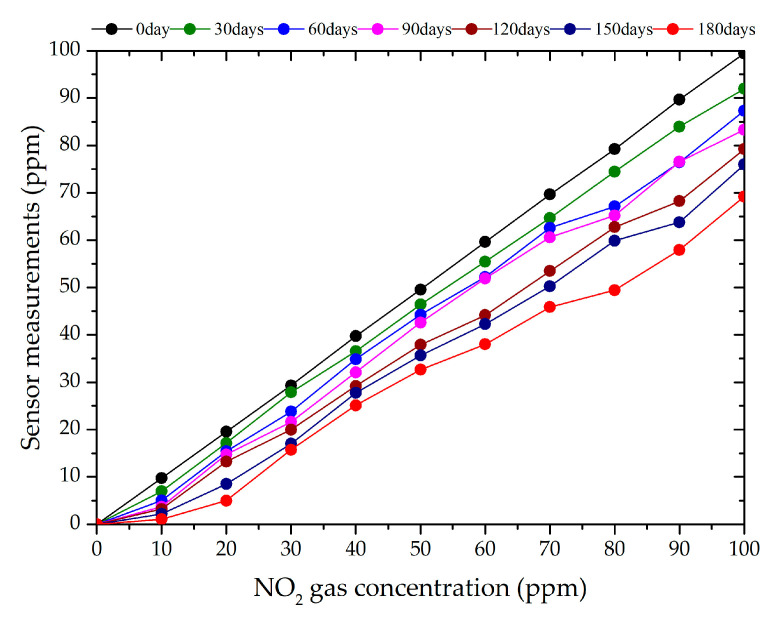
Experimental measuring of NO_2_ electrochemical sensors for different gas concentrations and deposition durations.

**Figure 11 sensors-24-00900-f011:**
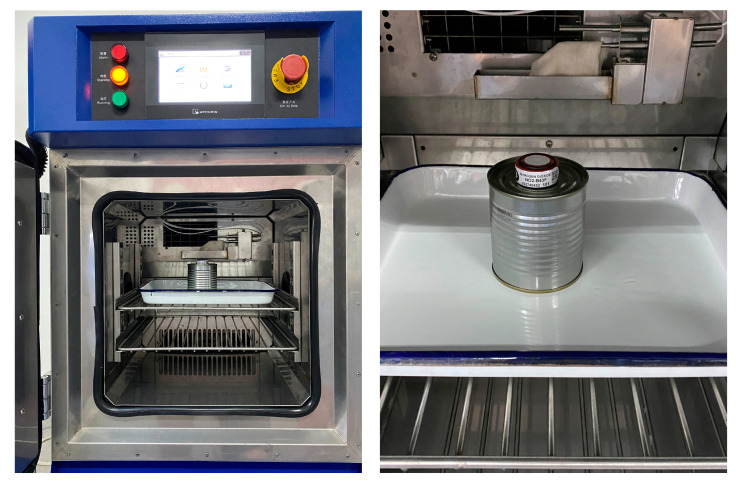
Control of humidity around NO_2_ electrochemical gas sensor.

**Figure 12 sensors-24-00900-f012:**
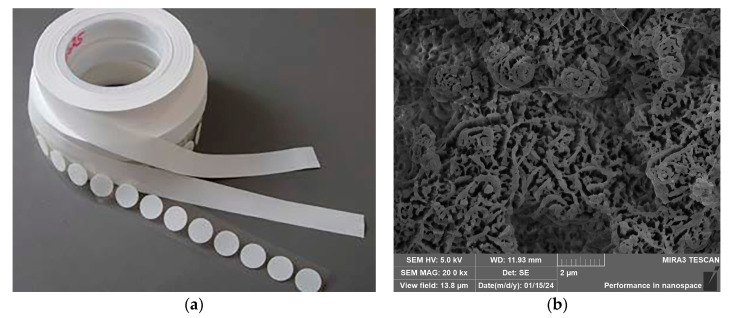
A filter membrane installed on the outside of a sensor’s equipment case. (**a**) PTFE membrane material fabricated by POREX; (**b**) SEM result showing morphological characteristics of the filter membrane with 2 μm scale bar.

**Table 1 sensors-24-00900-t001:** Performance parameters of the NO_2_ electrochemical gas sensor.

Category	Indicator	Description	Value
Specifications	Sensitivity	nA/ppm at 2 ppm NO_2_	−200 to −650
	Response time	t_90_(S) from 0 to 2 ppm NO_2_	<80
	Zero current	nA in zero air at 20 °C	−80 to +80
	Noise	±2 standard deviations (ppb equivalent)	15
	Range	ppm NO_2_ limit of performance warranty	20
	Linearity	ppb error at full scale, linear at zero and 20 ppm	<±0.5
	Overgas limit	NO_2_ maximum ppm for stable response to gas pulse	50
Lifetime	Zero drift	ppb equivalent change/year in lab air	0 to 20
	Sensitivity drift	% change/year in lab air, monthly test	−20 to −40
	Operating life	Months until 50% original signal (24-month warranty)	>24
Environmental	Sensitivity at −20 °C	% (output at −20 °C/output at 20 °C) at 2 ppm NO_2_	60 to 80
	Sensitivity at 40 °C	% (output at 50 °C/output at 20 °C) at 2 ppm NO_2_	95 to 115
	Zero at −20 °C	nA	0 to 25
	Zero at 40 °C	nA	−10 to 50
Cross-sensitivity	H_2_S	% measured gas at 5 ppm	<−80
	NO	% measured gas at 5 ppm	<5
	Cl_2_	% measured gas at 5 ppm	<100
	SO_2_	% measured gas at 5 ppm	<−3
	CO	% measured gas at 5 ppm	<−3
Key specifications	Temperature range	°C	−30 to 40
	Pressure range	kPa	80 to 120
	Humidity range	%rh continuous	15 to 85
	Storage period	Months at 3 to 20 °C	6
	Load resistor	Ω (ISB circuit is recommended)	33 to 100
	Weight	g	<13

Data from NO_2_-B43F electrochemical sensor instruction manual.

**Table 2 sensors-24-00900-t002:** The elemental composition of coal-fired fly ash (%).

O	Si	Al	C	Fe	K	Ca	Na
50.39	20.40	19.94	6.23	0.92	0.80	0.72	0.60

Data obtained from MIRA3 scanning.

**Table 3 sensors-24-00900-t003:** Pellet diameter distribution of coal-fired fly ash.

Distribution/μm	Average/μm	Quantity	Percentage/%
1–3	2.50	51	20.99
3–5	3.97	87	35.80
5–7	5.84	47	19.34
7–9	7.89	26	10.70
9–11	9.83	13	5.35
11–13	12.04	12	4.94
13–15	14.44	4	1.65
15–17	15.2	2	0.82
17–19	0	0	0.00
19–21	20.07	1	0.41

Data obtained from Nano Measurer software statistical results.

**Table 4 sensors-24-00900-t004:** Theoretical deposition quantities of coal-fired fly ash for various deposition durations.

Deposition Time	1 h	1 Day	30 Days	60 Days	90 Days	120 Days	150 Days	180 Days
Deposition Quantities/mg	33.1 × 10^−3^	0.80	24	48	72	96	120	144

Data from theoretical calculations.

**Table 5 sensors-24-00900-t005:** Experimental measuring of NO_2_ electrochemical sensors for different gas concentrations and deposition durations.

Sensor Measurement/ppm		Deposition Duration/Days						
		0	30	60	90	120	150	180
NO_2_ gas concentration/ppm	0	0	0	0	0	0	0	0
	10	9.740	6.965	5.020	3.682	3.222	2.182	1.051
	20	19.551	17.141	15.409	14.727	13.227	8.500	4.985
	30	29.318	27.869	23.788	21.591	19.975	17.015	15.742
	40	39.717	36.551	34.869	32.091	29.192	27.753	25.111
	50	49.568	46.409	44.232	42.586	37.894	35.646	32.641
	60	59.636	55.470	52.217	51.914	44.146	42.273	38.005
	70	69.646	64.662	59.596	60.591	53.475	50.278	45.859
	80	79.207	74.460	67.131	65.202	62.753	59.889	49.449
	90	89.712	83.965	76.439	76.576	68.242	63.773	57.939
	100	99.439	91.985	87.343	83.338	79.232	75.970	69.217

Data from experimental testing.

**Table 6 sensors-24-00900-t006:** Measurement errors of NO_2_ electrochemical sensors for different gas concentrations and deposition durations.

Sensor Measuring/ppm		Measurement Errors/%						
		0	30	60	90	120	150	180
NO_2_ gas concentration/ppm	0	0	0	0	0	0	0	0
	10	2.600	30.350	49.800	63.180	67.780	78.180	89.490
	20	2.245	14.295	22.955	26.365	33.865	57.500	75.075
	30	2.273	7.103	20.707	28.030	33.417	43.283	47.527
	40	0.708	8.623	12.828	19.773	27.020	30.618	37.223
	50	0.864	7.182	11.536	14.828	24.212	28.708	34.718
	60	0.607	7.550	12.972	13.477	26.423	29.545	36.658
	70	0.506	7.626	14.863	13.441	23.607	28.174	34.487
	80	0.991	6.925	16.086	18.498	21.559	25.139	38.189
	90	0.320	6.706	15.068	14.916	24.176	29.141	35.623
	100	0.561	8.015	12.657	16.662	20.768	24.030	30.783

Data from experimental testing.

**Table 7 sensors-24-00900-t007:** Experimental measurement of NO_2_ electrochemical sensors for different humidity levels.

Sensor Measurement/ppm		Humidity Level			
		25%RH	50%RH	75%RH	98%RH
NO_2_ gas concentration/ppm	0	0	0	0	0
	10	9.538	9.532	9.647	9.727
	20	19.251	19.407	19.725	19.891
	30	28.650	29.123	29.414	29.379
	40	38.571	38.786	39.025	39.449
	50	48.946	49.099	49.197	49.446
	60	58.818	58.197	59.167	59.217
	70	68.010	68.616	69.379	69.488
	80	78.227	79.069	79.118	79.808
	90	88.300	88.714	88.852	89.537
	100	98.906	98.498	99.655	99.828

Data obtained from experimental testing.

**Table 8 sensors-24-00900-t008:** Experimental measuring of NO_2_ electrochemical sensors at different humidity–coal fly ash deposition levels.

Sensor Measuring/ppm		30 Days’ Deposition				60 Days’ Deposition			
		25% RH	50% RH	75% RH	98% RH	25% RH	50% RH	75% RH	98% RH
NO_2_ gas concentration/ppm	0	0	0	0	0	0	0	0	0
	10	5.449	5.530	5.879	5.283	4.813	4.692	4.187	4.005
	20	16.641	15.884	14.995	14.611	15.874	15.162	15.010	14.616
	30	24.828	22.742	22.177	20.712	23.192	22.268	22.232	22.010
	40	36.273	35.439	33.970	28.571	34.470	33.505	33.064	32.793
	50	46.343	43.975	41.040	37.854	43.672	41.793	40.066	39.081
	60	53.485	51.389	49.601	49.551	51.899	50.465	49.141	48.626
	70	64.505	63.616	62.687	59.848	61.803	61.581	60.793	59.520
	80	72.525	71.641	71.066	67.268	66.768	65.742	65.414	64.404
	90	82.510	82.217	82.131	78.727	76.455	75.192	73.212	72.990
	100	90.404	89.722	87.616	84.869	84.030	83.566	83.535	83.071

Data from experimental testing.

## Data Availability

Data are contained within the article.

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
