# Peer review of "Influence of Coal-Fired Fly Ash on Measurement Error of NO2 Electrochemical Sensors"

_sensors, 2024, doi:10.3390/s24030900_

Round 1

Reviewer 1 Report

Comments and Suggestions for Authors

The manuscript entitled “Research of the Influence of Coal-Fired Fly Ash on the Measu ing Error of NO2 Electrochemical Sensors” has been submitted by authors. Some issues to be addressed which will improve the quality of manuscript. Therefore, I recommend this work could be published after the major revision

1)      In Introduction part, need to add new paragraph with comparative result of recent study.

2)      The experimental section is not clear and needs to be rewritten.

3)      There are some mistakes, so the language of the manuscript needs to be revised.

4)      Author need to incorporated gas sensing mechanism based on this study.

5)      For more accuracy to calculate error parameter, I recommend authors to compare gas sensing performance with other gases.

6)      The author must show the elemental mapping of the ratio of check materials contained in Coal-Fired Fly Ash.

7)      To enhance the strength of the manuscript and a broader readership range, some important references, needs to be incorporated as given below.

10.1021/acs.est.9b01725; 10.1016/j.jksus.2020.08.024

Comments on the Quality of English Language

Minor editing of English language required

Reviewer 2 Report

Comments and Suggestions for Authors

This work takes NO2 electrochemical sensors as its research object and, through experimental tests, investigates the influence of coal-fired fly ash deposition on their measuring error.  However, there are issues in the article. Therefore, minor revision is recommended.

1. Summarize the measurement errors of NO2 electrochemical sensors in other literature in the environment of coal ash.

2. Please explain the impact of different environments (such as temperature, humidity, etc.) on the measurement error of NO2 electrochemical sensors.

3. Please provide a detailed description of the solution to address the impact of coal ash environment on NO2 electrochemical sensors.

Comments on the Quality of English Language

Minor editing of English language required

Reviewer 3 Report

Comments and Suggestions for Authors

In this work, authors report on the measuring errors of NO2 electrochemical sensors that could be caused by the influence of the coal-fired fly ash. This is an interesting paper and the results are well supported by the data presented. However, my only objection is that the whole study is based on the performance of a specific commercial electrochemical NO2 sensor from the UK. Maybe the whole study would be more acceptable if a second commercial NO2 sensor was used for comparison and reproducibility purposes. I have the following comments for the authors.

1. The English in the whole manuscript is ok, however in general it could be improved, particularly in the experimental and results sections. For example, in section 2.2.2, page 6, lines 142-148 rather than giving instructions of how to place the electrochemical sensors etc., it would be better if the whole paragraph is written in the third person describing the experimental test steps.

2. In the introduction although the authors talk about two pollutant gases, NO and NO2 they decided to focus on the NO2 electrochemical sensors? Please justify your choice, are there any electrochemical sensors that could measure both gases at the same time? If yes, why the authors decided to base their study only on the NO2 gas?

3. Please state in the introduction what levels of NO­2 are harmful for human health, the authors talk about that without giving numbers. Except metal oxides and electrochemical methods, please mention more accurate analytical methods used for measuring these harmful gases.

4. Please name a few electrochemical commercial NO2 sensors and explain why you selected only one for this study.

5. In figure 10, on the X axis please leave space between concentration and units, also write number 2 as a subscript. For the units the same applies to the Y axis of the figure.

Comments on the Quality of English Language

The English in the whole manuscript is ok, however in general it could be improved, particularly in the experimental and results sections. For example, in section 2.2.2, page 6, lines 142-148 rather than giving instructions of how to place the electrochemical sensors etc., it would be better if the whole paragraph is written in the third person describing the experimental test steps.

Round 2

Reviewer 1 Report

Comments and Suggestions for Authors

The author answers all comments carefully. I recommend accepting the present form